# Not All Demonstration Examples are Equally Beneficial: Reweighting Demonstration Examples for In-Context Learning

**Zhe Yang, Damai Dai, Peiyi Wang, Zhifang Sui**
National Key Laboratory for Multimedia Information Processing,
School of Computer Science, Peking University
{yz_young,daidamai,szf}@pku.edu.cn
wangpeiyi9979@gmail.com

## Abstract

Large Language Models (LLMs) have recently gained the In-Context Learning (ICL) ability with the models scaling up, allowing them to quickly adapt to downstream tasks with only a few demonstration examples prepended in the input sequence. Nonetheless, the current practice of ICL treats all demonstration examples equally, which still warrants improvement, as the quality of examples is usually uneven. In this paper, we investigate how to determine approximately optimal weights for demonstration examples and how to apply them during ICL. To assess the quality of weights in the absence of additional validation data, we design a masked self-prediction (MSP) score that exhibits a strong correlation with the final ICL performance. To expedite the weight-searching process, we discretize the continuous weight space and adopt beam search. With approximately optimal weights obtained, we further propose two strategies to apply them to demonstrations at different model positions. Experimental results on 8 text classification tasks show that our approach outperforms conventional ICL by a large margin. Our code are publicly available at https://github.com/Zhe-Young/WICL.

## 1 Introduction

With the increase in model size and training corpus, Large Language Models (LLMs) have demonstrated in-context learning (ICL) capabilities (Radford et al., 2019; Brown et al., 2020; Wei et al., 2022; Dong et al., 2022). Unlike traditional fine-tuning, which requires updating model parameters, ICL allows LLMs to adapt to downstream tasks while keeping the parameters fixed. To enable ICL, a small number of examples will be prepended before the query to form a prompt. The prompt is then fed into the LLM for prediction. Numerous studies have shown the effectiveness of ICL, demonstrating that it can achieve, and sometimes even surpass, the performance of fine-tuned models.

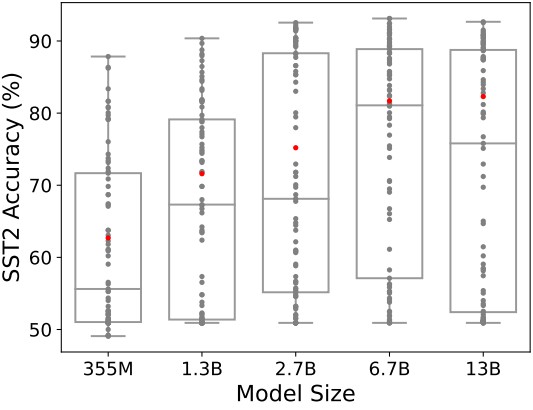

Figure 1: 4-shot ICL performance on SST2 with different sets of weights assigned to demonstration examples. Gray points represent the accuracy of different weight sets, and the red point denotes the accuracy of the non-weighting strategy. The performance varies significantly with different weights.

The performance of ICL is closely related to the quality of demonstration examples (Wu et al., 2022; Gonen et al., 2022). Therefore, various methods have been proposed to select high-quality examples. For example, Liu et al. (2022) selects nearest examples based on semantic similarity, Zhang et al. (2022) employs reinforcement learning for example selection, and Sorensen et al. (2022) maximizes mutual information on an unlabeled validation set for choosing templates. However, these methods assume the availability of a large number of examples for selection, which is not the case in real few-shot settings. From another aspect, previous ICL methods treat all demonstration examples equally, which can be further improved, as the quality of demonstrations is not even. As shown in Figure 1, assigning different sets of weights to demonstration examples has a significant impact on ICL performance. For a 13B GPT model, in terms of the accuracy on the SST2 dataset, the best weights surpass the worst by 40 points and surpass the non-

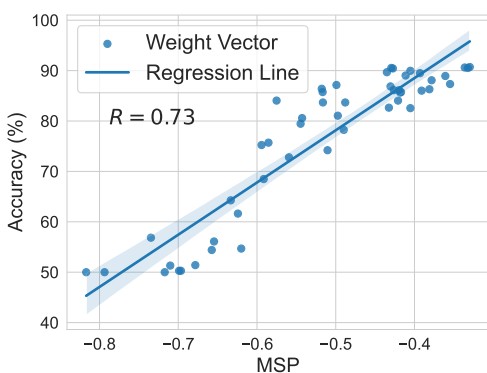

Figure 2: Regression line of MSP score and accuracy on MR dataset. Each point denotes the performance of a weight vector, the Pearson correlation coefficient is 0.73, indicating a strong correlation between MSP score and accuracy.

weighting strategy by 15 points. This highlights the necessity of assigning appropriate weights to demonstration examples.

In this paper, we propose the Weighted In-context Learning (WICL) method to enhance ICL by **determining** and **applying** approximately optimal weights to demonstration examples. There are two key challenges for determining weights: how to evaluate the quality of a set of weights without additional validation data (*weight evaluation*), and how to efficiently search the best set of weights in the continuous space (*weight search*). For weight evaluation, we introduce a Masked Self Prediction (MSP) score as a proxy metric, which can be computed solely on demonstration examples but is strongly correlated with the final ICL performance, as shown in Figure 2. For weight search, we discretize the weight space and employ beam search to efficiently find the weights that yield the highest MSP score. With approximately optimal weights discovered, we propose two strategies to apply them to demonstrations. Scaling Key Matrix (SKM) applies the weights to the attention key matrices of demonstration examples, while Scaling Attention Weights (SAW) directly adjusts the attention weights. Both strategies adjust the importance of demonstrations by influencing the attention weights according to a set of weights.

We evaluate WICL on 8 text classification tasks and show that it achieves substantial accuracy improvement over the conventional ICL method that follows a non-weighting strategy. In addition, our approach exhibits robustness to varying shot numbers and templates. As for the approximately optimal weights discovered by our approach, we find that they demonstrate a close performance to the global optimal weights. Furthermore, we discover that our example reweighting approach mainly works at middle layers, and reweighting only in a few middle layers even outperforms full layers.

We summarize our contributions as follows:

1. We propose the Weighted In-context Learning (WICL) method, which enhances ICL by determining and applying approximately optimal weights to demonstration examples.

2. We introduce a proxy metric called Masked Self-Prediction (MSP) score to foretell the final ICL performance without additional validation data.

3. We evaluate WICL on 8 NLP tasks and show that WICL can significantly outperform the conventional ICL method.

4. We perform elaborate quantitive analysis to reveal the robustness of WICL, and the quality of the discovered approximately optimal weights.

## 2 Problem Formulation

For a text classification task, we are given a set of $k$ demonstration examples denoted by $S = \{(x_1, y_1), (x_2, y_2), ..., (x_k, y_k)\}$, where $x_i$ and $y_i$ denote the input text and its corresponding label, respectively. We define a transformation function $T$ that maps each sample $(x, y)$ to an ICL example $T(x, y)$ (e.g. $T(x, y)$ = "Sentence: $x$ Sentiment: $y$" for the SST2 dataset). The concatenation of these transformed examples is then defined as the final demonstration $C$:

$$C = T(x_1, y_1) \oplus T(x_2, y_2) \oplus ... \oplus T(x_k, y_k), \quad (1)$$

where $\oplus$ denotes concatenation. The goal of ICL is to identify a suitable demonstration to enable the LLM to generate a probability distribution $P$ that closely approximates the actual distribution, and subsequently select the label $y_{predict}$ with the highest probability as the predicted output:

$$y_{predict} = \text{argmax}_{y \in D} P(y|C \oplus x; \theta) \quad (2)$$

In the above equation, $\theta$ and $D$ represent the parameters of LLM and the candidate label set,

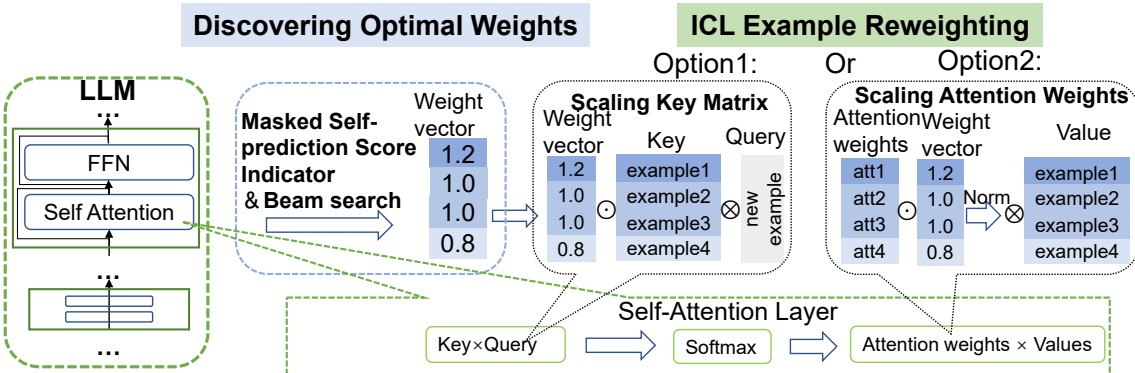

Figure 3: An illustration of weighted in-context learning. Reweighting at the self-attention layer could be scaling key matrix or scaling attention weights. The example weights can be obtained by beam search with masked self-prediction score as an indicator, which shows a strong correlation with final performance.

respectively. Conventional ICL does not take the weight of each example into account, and LLM almost treats each example equally. To address this limitation, we introduce a weight vector $w = (w_1, w_2, ..., w_k)$ for the demonstration, where $w_i$ denotes the weight of the $i^{th}$ example $T(x_i, y_i)$. A higher value of $w_i$ indicates that the example $T(x_i, y_i)$ is more important, and thus LLM should pay more attention to it. When given a specific demonstration, the objective of WICL is to fully use demonstration examples and identify an appropriate weight vector $w$ that enables the model to achieve performant performance.

## 3 Methodology

Demonstration example reweighting faces two challenges: (1) How to add weights to demonstration examples when given a weight vector. (2) How to find a performant weight vector in the absence of a held-out validation set. Section 3.1 aims to solve (1) by reweighting at the self-attention layer, and Section 3.2 aims to address (2) through weight quantization and beam search with an indicator. Our approach is illustrated in Figure 3.

### 3.1 Reweighting Demonstration Examples

Almost all of the current LLMs are based on Transformer decoder architecture, comprising multiple repeated blocks, each of which contains a multi-head self-attention layer and a feed-forward network layer. Self-attention layer is responsible for token interaction, and we can modify the weights for each example at this layer. Intuitively, increasing the attention weights that examples receive in the self-attention layer can be viewed as assigning more weights to the examples. Inspired by this,

we propose two simple yet effective approaches to assigning weights to examples by Scaling Key Matrix (SKM) or Scaling Attention Weights (SAW) respectively.

**Reweighting by Scaling Key Matrix** In self-attention layer, demonstration examples represent key, and new example to be predicted can be viewed as query. So we can add weights to demonstration examples by scaling vectors in the key matrix :

$$W_K = [W_{K,1}, W_{K,2}, ..., W_{K,k}], \qquad (3)$$

where $W_{K,i} \in \mathbf{R}^{d \times l_i}$ denotes the key matrix for the $i^{th}$ example, and $d$ and $l_i$ denote the hidden dimension and length of the $i$-th example, respectively. After scaling by $w$, the weighted key matrix can be represented as:

$$w \odot W_K = [w_1 W_{K,1}, ..., w_k W_{K,k}], \qquad (4)$$

and the weighted self-attention is calculated by the following equation:

$$\text{Attention}(W_V, w W_K, W_Q)$$
$$= W_V \text{softmax}(\frac{[w \odot W_K]^T W_Q}{\sqrt{d}}), \qquad (5)$$

where $W_V$ denotes value matrix; $W_Q$ denotes query matrix; $\sqrt{d}$ denotes scaling factor.

**Reweighting by Scaling Attention Weights** After the softmax layer, attention weights (the product query and key) are mapped into normalized probability distribution space. Adding weights to examples could be scaling attention weights of examples while maintaining the sum of them to be

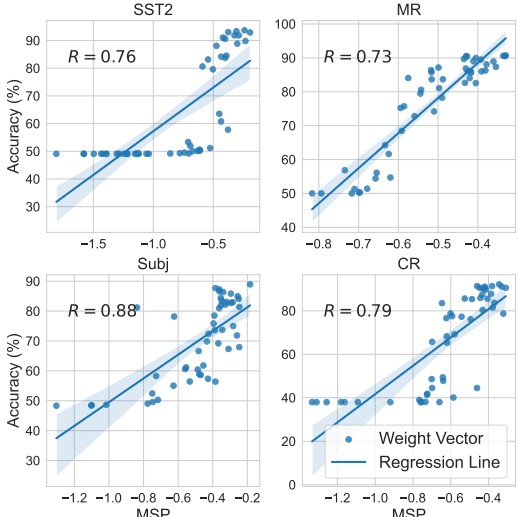

Figure 4: Correlation of MSP and accuracy under different example weights. For each task, we randomly sample 50 legal weights under 8-shot setting and test accuracy on GPT-1.3B, showing scatter plots and regression lines.

1. If attention weight for $i^{th}$ example is denoted as $att_i$, we have:

$$[att_1, ..., att_k] = \text{softmax}(\frac{W_K^T W_Q}{\sqrt{d}}) \quad (6)$$

We scale original attention weights $att$ by weight vector $w$ and normalize them. After scaling, new attention weight for $i^{th}$ example is :

$$att_i^{new} = \frac{w_i att_i}{\sum_{j=1,2,...,k} w_j att_j} \quad (7)$$

### 3.2 Discovering Optimal Weights

**Indicator for Weight Selection** How to select the optimal weight is a challenging problem. One naive approach is to test the model's performance on a validation set under a range of weights and select the weight that yields the best result. However, this approach may not be feasible when an extra validation set is unavailable. Fortunately, we have discovered a new metric called **average Masked Self-Prediction score (MSP)**, which can be obtained without the need for an additional validation set yet still has a strong correlation with the model's performance (as shown in Figure 4).

We consider the demonstration set $S$ as a special "validation set" and predict the example labels in $S$ conditioned on weight vector $w$. This process is

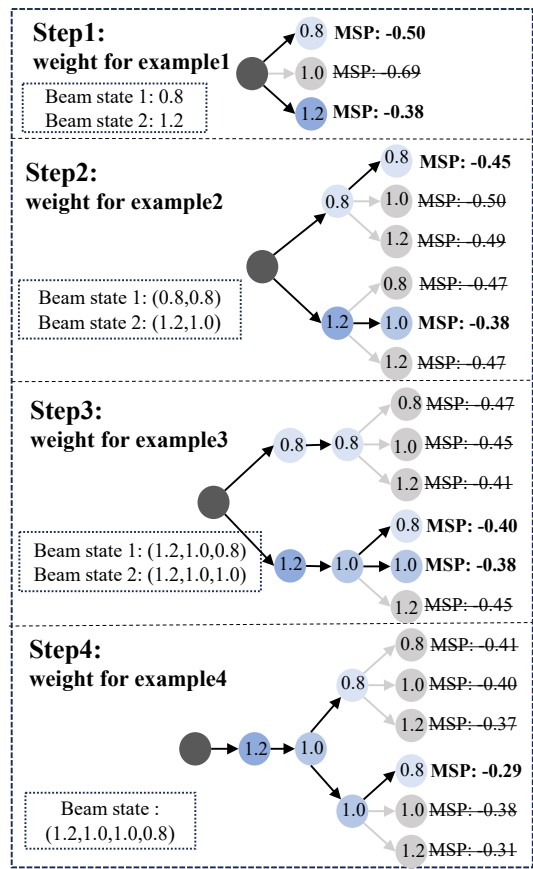

Figure 5: An illustration of beam search for example weights. We take 4-shot setting, beam size = 2 as an example, and legal weight set for each example is{0.8,1.0,1.2}. In each step, we extend beam states and preserve the 2 states with max MSP score.

called self-prediction since we are predicting the demonstration itself. Label $y_i$ in the demonstration $C$ is masked when predicting sample $(x_i, y_i)$ to prevent the model from copying the answers from the demonstration. MSP is defined as the average log probability of the predictions on $S$, and it is calculated as follows:

$$C_{mask-i} = ... \oplus T(x_i, [mask])... \oplus T(x_k, y_k) \quad (8)$$

$$p_i(w) = P(y_i | C_{mask-i} \oplus x_i; w, \theta) \quad (9)$$

$$MSP(w) = \frac{\sum_{i=1,2,...,k} log(p_i(w))}{k} \quad (10)$$

To find a performant weight, we can simply choose a weight with a high MSP score, instead of testing and comparing the performance of all weights on a validation set.

**Example Weight Searching** To select a proper weight vector $w \in R^k$, we employ **weight quantization strategy** that restricts each dimension $w_i$

| | Model | Method | SST2 | CR | AGN | RTE | TREC | DBP | MR | Subj | Avg |
|---|---|---|---|---|---|---|---|---|---|---|---|
| **8 shot** | GPT 335M | ICL | 72.1 | 65.7 | 47.0 | 53.7 | 34.5 | 75.0 | 65.7 | 49.7 | 57.9 |
| | | WICL-SAW | 77.1 | 76.4 | 54.6 | 54.1 | **36.1** | **76.3** | 71.1 | 54.5 | 62.5 |
| | | WICL-SKM | **80.9** | **81.1** | **55.4** | **54.3** | 34.7 | 75.0 | **75.5** | **59.9** | **64.6** |
| | GPT 1.3B | ICL | 64.9 | 51.0 | 74.8 | 53.2 | 32.2 | 61.3 | 71.2 | 64.0 | 59.1 |
| | | WICL-SAW | 68.7 | 55.4 | 76.3 | 53.2 | 33.3 | 64.1 | 71.1 | **69.3** | 61.4 |
| | | WICL-SKM | **76.5** | **74.9** | **78.5** | **53.3** | **33.7** | **68.0** | **73.4** | 69.0 | **65.9** |
| | GPT 2.7B | ICL | 68.1 | 63.1 | 79.1 | 53.0 | 27.1 | 86.1 | 83.8 | 66.4 | 65.8 |
| | | WICL-SAW | 71.6 | 72.2 | **80.1** | **53.9** | 28.4 | 86.1 | 86.4 | 71.8 | 68.8 |
| | | WICL-SKM | **79.1** | **80.2** | 79.7 | 53.7 | **29.4** | **86.7** | **87.8** | **75.6** | **71.5** |
| | GPT 6.7B | ICL | 87.4 | 73.0 | 78.7 | 57.1 | 28.0 | 85.4 | 82.8 | 52.5 | 68.1 |
| | | WICL-SAW | **88.5** | 74.8 | 79.7 | **57.5** | 28.6 | **85.8** | 84.6 | 53.3 | 69.1 |
| | | WICL-SKM | **88.5** | **82.0** | **82.1** | 57.3 | **29.3** | 85.2 | **85.2** | **57.3** | **70.9** |
| | GPT 13B | ICL | 89.5 | 87.0 | 85.3 | 51.9 | 34.0 | 87.9 | 88.2 | 58.8 | 72.8 |
| | | WICL-SAW | **90.6** | 88.1 | 85.4 | 53.2 | **34.3** | 88.1 | **89.5** | 60.6 | 73.7 |
| | | WICL-SKM | 89.4 | **88.9** | **85.6** | **57.8** | 33.8 | **88.2** | 88.8 | **68.2** | **75.1** |
| **16 shot** | GPT 335M | ICL | 64.4 | 66.9 | 44.3 | 53.3 | 38.9 | 72.7 | 68.8 | 49.2 | 57.3 |
| | | WICL-SAW | 78.5 | 77.3 | **56.7** | 53.2 | **41.4** | 74.8 | 71.4 | 56.2 | 63.7 |
| | | WICL-SKM | **84.0** | **79.9** | 55.0 | **54.3** | 40.6 | **75.1** | **75.9** | **61.6** | **65.8** |
| | GPT 1.3B | ICL | 65.0 | 58.1 | 75.0 | **53.6** | 35.2 | 64.5 | 72.3 | 64.6 | 61.0 |
| | | WICL-SAW | 72.1 | 66.0 | 77.2 | 53.2 | **37.6** | 68.3 | **74.0** | **75.1** | 65.4 |
| | | WICL-SKM | **82.6** | **76.0** | **78.7** | 53.5 | 36.6 | **73.9** | 73.5 | 74.2 | **68.6** |
| | GPT 2.7B | ICL | 69.4 | 60.7 | 79.9 | 53.0 | 27.7 | 87.0 | 82.6 | 62.2 | 65.3 |
| | | WICL-SAW | 78.1 | 76.5 | **80.1** | **53.6** | 29.8 | 87.2 | 87.2 | 74.6 | 70.9 |
| | | WICL-SKM | **83.8** | **83.9** | 79.5 | 53.5 | **31.4** | **87.3** | **88.3** | **81.9** | **73.7** |
| | GPT 6.7B | ICL | 88.4 | 60.9 | 78.9 | 57.4 | 28.9 | **86.6** | 86.7 | 52.0 | 67.5 |
| | | WICL-SAW | 89.9 | 65.0 | 80.1 | **57.9** | 31.4 | **86.6** | **87.7** | 52.4 | 68.9 |
| | | WICL-SKM | 89.7 | **81.9** | **81.2** | 57.8 | **31.7** | 85.8 | 87.1 | **64.7** | **72.5** |
| | GPT 13B | ICL | 90.7 | 83.0 | 85.4 | 51.1 | 34.6 | 89.3 | 87.8 | 61.9 | 73.0 |
| | | WICL-SAW | **92.0** | 85.9 | **85.7** | 53.2 | **37.0** | **89.4** | 89.7 | 64.5 | 74.7 |
| | | WICL-SKM | 91.2 | **89.7** | 85.7 | **58.7** | 37.0 | 89.2 | **89.8** | **77.2** | **77.3** |

Table 1: Main experiment results. Taking conventional ICL as a baseline, we compare the performance of WICL with two reweighting methods on 8 datasets with different models under 8-shot and 16-shot settings. For simplicity, DBPedia and AGNews are written as DBP and AGN, respectively.

in $w$ to values in a candidate weight set $Q = \{weight_1, weight_2, ..., weight_n\}$. This strategy compresses weights from continuous infinite space to discrete finite space and the number of legal weights drops from infinite to $n^k$, making the weight selection problem computationally tractable. However, brute force enumeration of all of the possible weights and calculation of MSP score still requires $O(k * n^k)$ time complexity, which is unscalable for large $k$. To address this problem, we apply **beam search strategy** which is commonly used in language model decoding process. From example 1 to example k, in each beam search step we search for a weight for one example, as illustrated in Figure 5. The pseudo-code of our weight selection approach is presented in Appendix A.

## 4 Experiments

Similar to previous work (Lu et al., 2022), we use 8 classical text classification datasets, respectively SST2 (Socher et al., 2013), CR (Hu and Liu, 2004), MR (Pang and Lee, 2005), Subj (Pang and Lee, 2004), TREC (Voorhees and Tice, 2000), DBPedia (Zhang et al., 2015), AGNews (Zhang et al., 2015) and RTE (Dagan et al., 2006), to evaluate the efficacy of our demonstration examples reweighting approach. In our experiments, we utilize 5 different GPT-like causal language models released by fairseq (Artetxe et al., 2022), and the number of parameters of these models is 355M, 1.3B, 2.7B, 6.7B, 13B, respectively.[1]

Previous works (Wu et al., 2022; Gonen et al., 2022; Liu et al., 2022; Zhang et al., 2022) have

---

[1] https://github.com/facebookresearch/fairseq/tree/main/examples/moe_lm

|            | 355M | 1.3B | 2.7B | Avg  |
|------------|------|------|------|------|
| ICL Baseline | 72.1 | 64.9 | 68.1 | 68.4 |
| WICL-Dual  | 80.8 | 76.1 | 77.7 | 78.2 |
| WICL-SKM   | **80.9** | **76.5** | **79.1** | **78.8** |

Table 2: SST2 accuracy (%) for different methods and models. WICL-Dual denotes WICL by combining scaling key matrix and scaling attention weights. WICL-SKM outperforms WICL-Dual.

|            | 355M | 1.3B | 2.7B | Avg  |
|------------|------|------|------|------|
| ICL Baseline | 72.1 | 64.9 | 68.1 | 68.4 |
| Label-Only Masking | **80.8** | **76.5** | 79.1 | **78.8** |
| Whole-Example Masking | 75.0 | 74.8 | **81.2** | 77.0 |
| Whole-Example Removing | 59.0 | 70.0 | 63.4 | 64.1 |

Table 3: Comparing different approaches to calculate MSP on SST2. Label-only Masking strategy outperforms other strategies.

reported ICL performance depends heavily on the example selection, which is important but not the focus of our work. In order to reduce the possible effects of example selection on our experiments, we repeat each group of experiments 100 times under different seeds and report average results. For each group of experiments, we randomly sample demonstration examples from the training set with random seeds of 0,1,2, ... 99 respectively, and use balanced sampling strategy to ensure that labels of examples are as balanced as possible (e.g., 3,3,2 examples for different classes for a 3-classification task in 8-shot setting). To reduce computational cost, we limit the number of data of validation set to no more than 2000; for validation set with more than 2000 data, we only randomly sample 2000 data for testing. The experiments are conducted under 8-shot and 16-shot settings. The templates used in the experiments and more details are shown in Appendix B.

As the experiment results shown in Table 1, the average accuracy is reported under the following three methods: (1) conventional ICL, (2) WICL by scaling key matrix and (3) WICL by scaling attention weights. our approaches outperform conventional ICL on almost all 8 tasks, 5 different models. Besides, SKM also shows more power in performance improvement than SAW. We also try to combine these two reweighting strategies, which means reweighting by scaling key matrix and attention weights simultaneously. As shown in Table 2, combining SAW and SKM does not improve performance; instead, it does harm to the performance and the final performance is a little weaker than SKM.

## 5 Analysis

We also further analyze our experiments and investigate properties of our approach. Since SKM outperforms SAW, we do our analysis experiments only with SKM reweighting strategy. Firstly, we compare different example masking/removing

strategies when calculating MSP (Section 5.1), and compare MSP with held-out validation set (Section 5.2). Nextly, we explore the robustness of our approach to different templates, shot numbers (Section 5.3, 5.4). Then, we find that our approach can obtain near-optimal weights (Section 5.5). In Section 5.6, we discover that example reweighting mainly works at middle layers, and reweighting on only a few middle layers can even outperform full-layer reweighting. Section 5.7 shows some empirical results on relationship between example Weight and example quality/position.

### 5.1 Label-Only Masking Outperforms Whole-Example Masking/Removing

Our MSP score calculation process is similar to Leave-One-Out Cross-Validation (LOOCV) because we both test one example based on the other $k - 1$ examples when given k examples. However, unlike LOOCV, we only mask the label of the example to be tested rather than removing it entirely. Inspired by LOOCV, we also compare the performance of label-only making, whole-example masking and whole-example removing. As shown in Table 3, label-only masking outperforms other methods. Whole-example masking/removing results in a performance drop due to the demonstration sensitivity of ICL. When the entire example is masked/removed in MSP calculation, the corrupted demonstration differs significantly from the original one used in final testing, leading to inconsistency that harms ICL performance. Label-only masking can strike a balance between preserving demonstration information and preventing the model from copying the answers from the demonstration.

### 5.2 MSP is Approximation of a Held-out Validation Set

A held-out validation set can be viewed as a "strong supervision", while our MSP is more like an approximation of this supervision under true few-shot

|         | ICL  | WICL -MSP | WICL-Validation | | | |
|---------|------|-----------|------|------|------|------|
|         |      |           | 10   | 20   | 50   | 100  |
| 8-shot  | 64.9 | 76.5      | 76.1 | 82.4 | 83.2 | **84.9** |
| 16-shot | 65.0 | 82.6      | 79.6 | 83.0 | 83.4 | **85.3** |

Table 4: Comparing MSP with a held-out validation set of 10, 20, 50, 100 examples on SST2 using GPT 1.3B. Though validation set outperforms MSP, MSP can improve ICL significantly under true few-shot setting.

Figure 6: Performance of WICL on SST2 under different shot settings. WICL is robust to different shot numbers.

setting. The comparison of MSP and a held-out validation of 10, 20, 50, 100 examples are shown in Table 4. Though MSP is slightly weaker than validation set, it outperforms ICL baseline by a large margin. In true few-shot setting where resource is limited to only a few examples, MSP is an efficient indicator for weight search.

### 5.3 Example Reweighting is Robust under Different Shot Settings

Our main experiment is conducted on 8 and 16 shot settings, and we aim to explore the effects of example reweighting under different shot settings, as the experimental results presented in Figure 6. Our findings indicate that our approach does not significantly improve model performance when the number of shots is less than 4; however, when the number of shots is greater than 4, there is a significant improvement over the baseline. Moreover, we observe that the larger the number of shots, the model's ICL performance is not necessarily better, and the ICL performance may gradually decline with the increase of shots. This could be due to information distraction caused by too many examples. On the contrary, our approach can steadily improve

| Size | Method | Template | | | |
|------|--------|------|------|------|------|
|      |        | 1    | 2    | 3    | 4    |
| 355M | ICL    | 72.1 | 69.6 | 54.6 | 71.8 |
|      | WICL   | **80.9** | **71.5** | **69.2** | **80.0** |
| 1.3B | ICL    | 64.9 | 63.1 | 55.6 | 68.1 |
|      | WICL   | **76.5** | **79.3** | **73.8** | **80.6** |
| 2.7B | ICL    | 68.1 | 78.6 | 68.9 | 74.7 |
|      | WICL   | **79.1** | **85.3** | **83.1** | **85.5** |

Table 5: Performance of different models on SST2 under 4 different templates. WICL is robust to different templates.

with the increase of shot number, which demonstrates that example reweighting can enhance the model's ability to make full use of examples. We also observe that for SST2 (a binary classification task), the ICL performance fluctuates between odd and even shot numbers as the number of shots increases. Differently, this oscillation on WICL is relatively small, indicating that WICL is more robust to odd or even shot numbers.

### 5.4 Example Reweighting is Robust to Different Templates

In our main experiment, only one template is used for each task, and we would like to further explore the robustness of our method to different templates. As shown in Table 5, we use 4 different templates on the SST2 dataset, and for each template, our method can achieve a significant improvement over the conventional ICL method. Details of these templates are shown in Appendix C. Moreover, conventional ICL is sensitive to the choice of templates, and the performance can vary significantly under different templates (e.g. 17.2% difference in accuracy under template3 and template4 for GPT 355M). On the contrary, our approach significantly reduces the model's sensitivity to templates and performs well on different templates.

### 5.5 MSP and Beam Search Yields Near-optimal Weights

Finding the optimal example weights can be very expensive as it requires enumerating all legal weights and validating them. In contrast, our approach requires no extra data and has relatively low computational complexity. Therefore, it is worth exploring whether our simplification for obtaining weights significantly impacts performance and how far our example weights are from optimal weights. To investigate this, we randomly sample 50 weights

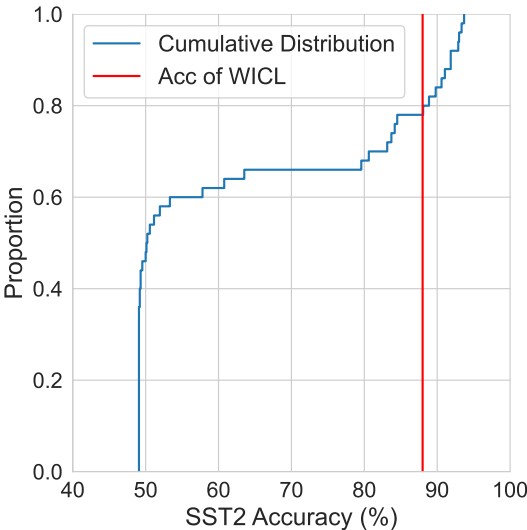

Figure 7: Cumulative distribution of accuracy under different weights on SST2. Our approach outperforms about 80% proportion of weights.

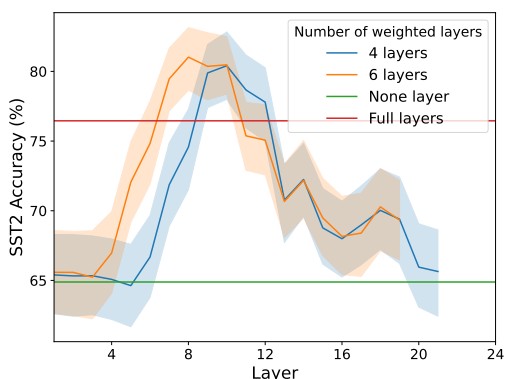

Figure 8: WICL performance of GPT 1.3B when only reweighting a few consecutive layers. Reweighting mainly work at middle layers.

from the weight space under the 8-shot setting, test the accuracy on the validation set, and plot the cumulative distribution function (as shown in Figure 7). Our findings indicate that our approach outperforms approximately 80% of the weights, demonstrating that the MSP indicator and beam search can yield near-optimal weights.

### 5.6 Example Reweighting Mainly Works at Middle Layers

In our main experiment, example reweighting is applied at all layers, but it is still unclear at which layers the reweighting strategy is most effective. To investigate this, we use a partial reweighting approach, where examples are reweighted only on certain layers of the model. we adopt direct ICL and

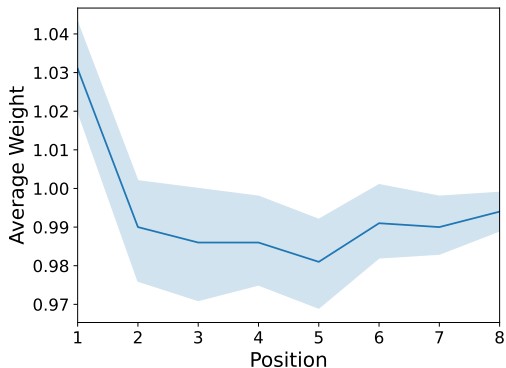

Figure 9: Average weight for examples in different positions for 8-shot WICL. Examples positioned in the middle tend to be assigned less weight, while those at the beginning are assigned more.

full-layer WICL as two baselines and compare the performance of reweighting only on a few consecutive layers, as shown in Figure 8. Our findings suggest that example reweighting has almost no effect at the top and bottom layers, but has a significant impact at the middle layers. Moreover, reweighting only a few intermediate layers can even outperform reweighting all layers.

### 5.7 Relationship between Example Weight and Example Quality/Position

As shown in Figure 9, we empirically find that weights of examples at different positions are not evenly distributed; examples positioned in the middle tend to be assigned less weight, while those at the beginning are assigned more. Besides, it has been observed that examples with higher 1-shot performance tend to maintain their original weights, while examples of worse 1-shot performance tend to be rescaled. For examples of weight 0.9, 1.0, and 1.1, their average 1-shot performance on SST2 is 61.4%, 65.2% and 58.3%, respectively. Example reweighting makes the demonstration strike an intrinsic balance by scaling these examples, and 1-shot performance indicates the intrinsic balance degree of the example.

## 6 Related Work

**In-Context Learning** The emergence of LLMs' in-context learning ability has drawn a fair amount of attention to the study of few-shot learning with LLMs. However, ICL is still faced with the problem of instability and vulnerability (i.e. huge differences in performance under different

prompts). Many existing works have focused on addressing this problem and improving ICL performance: Sorensen et al. (2022) attributes the poor performance to bad prompt section and proposes an information-theoretic approach to select prompt based on mutual information, which can be obtained with an external unlabeled set. Lu et al. (2022) considers the poor performance is from poor order of demonstration examples and proposes an order selection algorithm based on entropy, which can be obtained with the help of probing set produced by LLM itself. Zhao et al. (2021) proposes that the failure of ICL is mainly caused by all kinds of biases in demonstration examples or LLM itself, and designs a simple debias method. Different from previous work, we propose a new perspective that poor performance of ICL comes from lack of proper weights for demonstration examples and design an effective reweighting methodology for ICL.

**Reweighting Training Examples in Machine Learning**  Machine learning can easily overfit to training set biases, especially when there is a deviation between training set distribution and true distribution. To address this problem, reweighting training examples is a commonly used strategy. To reduce the variance of estimation, Importance Sampling (Kahn and Marshall, 1953) equalizes the two distributions by assigning weights to examples. AdaBoost (Freund and Schapire, 1997) iteratively assigns more weights to misclassified examples to obtain a stronger classifier; Focal Loss (Lin et al., 2017) assigns more weights to harder examples with a soft weighting scheme; Ren et al. (2018) learns to assign weights to training examples based on their gradient directions. However, previous example reweighting work mainly focuses on machine learning with abundant training data and few-shot learning is not included. Different from previous work, we try to explore example reweighting of ICL under few-shot setting.

**True Few-shot Learning**  In true few-shot setting, a held-out validation set for better performance is unavailable, and resource is limited to only a few examples. Previous work (Perez et al., 2021; Schick and Schütze, 2022; Bragg et al., 2021) have fully discussed the importance of true few-shot setting for ICL. Min et al. (2022a); Lu et al. (2022); Lyu et al. (2023) also try to improve ICL

under this setting. Our approach for ICL examples reweighting is under true few-shot setting.

## 7 Conclusion

In this paper, we find that treating demonstration examples equally can be a bottleneck for ICL, and assigning proper weights to examples can significantly improve performance. Based on this observation, we attempt to reweight demonstration examples and propose weighted in-context learning. To explicitly assign weights, we reweight examples in the self-attention layer by scaling key matrix or scaling attention weights; to determine the weight vector for examples, we adopt weight quantization beam search by maximizing masked self-prediction score, which can be obtained with any held-out validation set yet shows a strong correlation with the final performance. Our approach can significantly improve ICL performance in true few-shot setting, and experiments on 8 NLP tasks also demonstrate the efficacy of our approach.

## Limitations

Although our approach can significantly improve the performance of ICL by reweighting the demonstration examples, there are still many limitations: **(1)** Our approach only has a noticeable effect when the number of examples is greater than 4. **(2)** Under our approach, performance variance still exists and the quality of examples still matters to the performance. Our approach can improve performance but can not totally solve ICL's vulnerability to demonstration examples. **(3)** Our approach performs reweighting at the self-attention layer, which requires access to the model's parameters; for some large models whose parameters are unavailable (e.g. GPT4 (OpenAI, 2023)), our approach can hardly be applied. **(4)** We simply add the same weights to all layers or a few consecutive layers, without performing more fine-grained reweighting strategies to different layers (e.g. different example weights for different layers), and the mechanism of example reweighting on the improvement of ICL is still unexplored.

## Acknowledgements

This paper is supported by the National Key Research and Development Program of China 2020AAA0106700 and NSFC project U19A2065.

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

# Appendix

## A Algorithm for Weight Selection with MSP Score

---
**Algorithm 1: Weight Selection with MSP**

---
  **Input:**
  $Q$: candidate weight set for each example;
  $b$: beam size; $k$ shot number
  $f$: function to calculate MSP score;
  **Output:** weight vector $w$

**1 Initialize** $W_0 \leftarrow \{(1.0, 1.0, ..., 1.0)\}$
**2 for** $i \in \{1, 2, ..., k\}$ **do**
**3**    $W_i \leftarrow \emptyset$
**4**    **for** $w$ *in* $W_{i-1}$ **do**
**5**       **for** *candi in* $Q$ **do**
**6**          w[i] $\leftarrow$ candi
**7**          score $\leftarrow f(w)$
**8**          $W_i$.append( (score,w) )
**9**       **end**
**10**    **end**
**11**    $W_i \leftarrow W_i.topb()$
**12 end**
**13 Return** $W_k.top1()$

---

## B Experiment Details

For GPT-355M, GPT-1.3B, GPT-2.7B and GPT-6.7B, we do the experiments on a single NVIDIA 3090 24G GPU. For GPT-13B, we do the experiments on a single NVIDIA A40 48G GPU, because this model is too large to load on an NVIDIA 3090 24G GPU. For scaling key matrix approach, the weight candidate set is $\{0.9, 1.0, 1.1\}$, and for scaling attention weight approach, the weight candidate set is $\{0.8, 1.0, 1.2\}$. In beam search, we set beam size to 1 (i.e. greedy search). The template and label mapping in our experiments is shown in Table 6.

## C 4 Templates for SST2

To explore the robustness of WICL to different templates, we do the experiments on SST2 with 4 different templates. The templates and label mapping are shown in Table 7

| Tasks | Template | Label Mapping |
|---|---|---|
| SST2 | Sentence: {text} Sentiment: {answer} | negative, positive |
| CR | Sentence: {text} Sentiment: {answer} | negative, positive |
| AGNews | Article: {text} Category: {answer} | World,Sports,Business,Technology |
| RTE | {sentence1} Question: {sentence2} True or False? Answer: {answer} | True,False |
| TREC | Question: {text} Type: {answer} | Description, Entity, Expression, Person, Number, Location |
| DBPedia | Input: {content} Type: {answer} | company, school, artist, sport, politics, transportation, building, nature, village, animal, plant, album, film, book |
| MR | Review: {text} Sentiment: {answer} | Negative, Positive |
| Subj | Input: {text} Type: {answer} | objective, subjective |

Table 6: Templates and label mapping for different tasks.

| ID | Template | Label Mapping |
|---|---|---|
| 1 | Sentence: {text} Sentiment: {answer} | negative, positive |
| 2 | Input: {text} Prediction: {answer} | negative, positive |
| 3 | Review: {text} It was {answer} | bad, good |
| 4 | Review: {text} Sentiment: {answer} | bad, good |

Table 7: 4 different templates and label mapping for SST2.