# OpenReview forum: "Not All Demonstration Examples are Equally Beneficial: Reweighting Demonstration Examples for In-Context Learning"
_EMNLP/2023/Conference — EMNLP 2023 Findings_

### Official Review · Reviewer_zns4 · 2023-07-20

**Soundness:** 4

**Excitement:**

4: Strong: This paper deepens the understanding of some phenomenon or lowers the barriers to an existing research direction.

**Paper Topic And Main Contributions:**

This paper studies the weights of different demonstrations in in-context learning. They design a score to evaluate the quality of demonstration weights and search for approximately optimal weights to improve the final in-context learning performance. The contribution is ``approaches for data and compute efficiency''.

**Questions For The Authors:**

1. What are the weights look like found by WICL? Are there any patterns that exist in the examples if the method assigns high/low weights?
2. How does the order of the demonstrations influence the assigned weights?
3. The method seems more effective for small models than large models: for 8-shot, +6.7 on 335M, but +2.3 on 13B; for 16-shot, +8.5 on 335M but +4.3 on 13B. Following this tendency, will the improvement vanish on 70B models? (Of course, the improvement on the 13B model is also meaningful in practice. You can only share your thoughts if the resource is limited for larger experiments.)

**Reasons To Accept:**

1. The method proposed in the paper is simple and easy to follow.
2. The MSP score seems to be a good metric to evaluate the in-context learning performance under more circumstances.
3. The improvement of the weight searching method is constant across different datasets and model sizes.
4. The analysis provides valuable insights on the reweighting approach.

**Reasons To Reject:**

1. The experiments are mostly conducted on text classification tasks. How the method works on the text generation tasks, which are more used in current LLMs, is not clear.
2. The method introduces extra inference costs on LLMs when searching for the weights.


**Reproducibility:**

4: Could mostly reproduce the results, but there may be some variation because of sample variance or minor variations in their interpretation of the protocol or method.

**Reviewer Confidence:**

4: Quite sure. I tried to check the important points carefully. It's unlikely, though conceivable, that I missed something that should affect my ratings.

---

> ### Author Rebuttal · Authors · 2023-08-28
>
> # Responses to Reviewer zns4
> Thank you for taking the time to thoroughly review our paper, and I sincerely appreciate your feedback. Some of your questions are helpful to further improvement our paper. I’d like to answer your questions.
>
> ***
>
> ## Q1:
> “How the method works on the text generation tasks”
>
> ## Answer:
> Text classification is a special case of text generation when only one token/label is generated. Our approach can be easily applied to text generation task by replacing the likelihood of labels by PPL.
>
> ***
>
> ## Q2:
> "The method introduces extra inference costs on LLMs when searching for the weights."
>
> ## Answer:
> Computational complexity of our approach is not high at all.
> 1.The additional computation introduced by our method is a one-time cost, and we only need to search for proper weights one time before inference.
> 2.	Beam search is linear in time complexity for the number of shots. Even for 16-shot setting, searching for weights only needs 3*16 forward passes through language model.
> So the computational cost of our approach is totally acceptable.
>
> ***
>
> ## Q3:
>  "Are there any patterns that exist in the examples if the method assigns high/low weights?”, “How does the order of the demonstrations influence the assigned weights?”
>
>
> ## Answer:
> There is a certain correlation between weights and the positioning of examples, as well as the quality of examples. Examples positioned in the middle tend to be assigned less weight, while those at the beginning and end are assigned more. Besides, experiment result also shows higher-quality examples tend to be assigned more weight, which is consistent with intuition. That’s a good question, and we will add this part to section 5.
>
> ***
>
> ## Q4:
> “Will the improvement vanish on 70B models? ”
>
> ## Answer:
>
> In my view, the current research on enhancing ICL stems from the weaknesses inherent in the models themselves and their lack of robustness (e.g. sensitivity to examples and order). Compared to smaller models, larger models exhibit superior performance and greater robustness. Therefore, applying our method to a more powerful 70B model will result in a smaller improvement. However, this improvement won't vanish, as even the 70B model's ICL capabilities are not yet perfect. From another perspective, our approach is an economical and straightforward method that to some extent narrows the ICL performance gap between large and small models.

---

### Official Review · Reviewer_Mxfy · 2023-07-31

**Soundness:** 2

**Excitement:**

2: Mediocre: This paper makes marginal contributions (vs non-contemporaneous work), so I would rather not see it in the conference.

**Paper Topic And Main Contributions:**

This work proposes an example selection strategy for in-context learning. The authors propose to ‘weight’ the examples so that more relevant examples receive higher weight. The actual weighting is performed by rescaling the attention weights of the pre-trained language model. In order to find the optimal weights, a scoring mechanism is proposed to quantify the quality of a given weight vector which is based on the likelihood prediction of the label for a heldout example given demonstration examples. The optimal weights are chosen using a beam-search decoding procedure, where the weights are limited to a pre-defined set of discrete options. Experiments on 8 classification tasks show that the method performs better than baselines.

**Questions For The Authors:**

A. line 051 posits that prior example selection strategies need a lot of data to work. However, this is not empirically validated. I would suggest still trying them out and showing the empirical performance. Also, I don’t see why the leave-one-out approach used in the proposed method cannot be used with these baselines.

B. Is the search algorithm proposed sensitive to example order?

C. How is the discretized weight space chosen?

**Reasons To Accept:**

* Re-scaling the attention based on learned weights is an interesting idea.
* Ablations presented are somewhat useful.

**Reasons To Reject:**

The paper does not consider strong baselines. For instance, a simple baseline that could have been considered is retrieval/ranking based on query similarity.

This also makes me question whether the complexity of the method proposed is justified by the experimental results. The search algorithm proposed is computationally expensive and involves multiple forward passes through the language model.

In fig 6 a monotonic decrease is observed for the baseline ICL when number of demonstrations are increased, which is unusual. I suggest analyzing this behavior with more models and debugging the baseline implementation.

**Reproducibility:**

3: Could reproduce the results with some difficulty. The settings of parameters are underspecified or subjectively determined; the training/evaluation data are not widely available.

**Reviewer Confidence:**

3: Pretty sure, but there's a chance I missed something. Although I have a good feel for this area in general, I did not carefully check the paper's details, e.g., the math, experimental design, or novelty.

**Typos Grammar Style And Presentation Improvements:**

In section 3, the paper directly talks about query, key, value matrices without providing any background or motivation. As a result, I wasn’t able to get a complete understanding of the technical details.

I couldn’t get the main intuitions from Fig 3. The figure needs to be improved.

line 284: the description about random seeds isn’t very meaningful as random seeds are generally platform specific.

---

> ### Author Rebuttal · Authors · 2023-08-28
>
> # Resopnses to Reviewer Mxfy
>
> Thank you for taking the time to review our paper. However, it seems there is a great misunderstanding about core idea of our work. With regard to "this work proposes an example selection strategy for in-context learning", that’s not what our work actually does. Our work aims to find proper weight for examples in a given demonstration, and our work does not involve example selection, as selecting examples from a dataset contradicts true few-shot setting. We do experiments with randomly sampled examples, and our approach can improve ICL performance for almost any given demonstration under true few-shot setting.
>
> In **true few-shot setting**, a held-out validation set for better performance is unavailable, and resource is limited to only a few examples. Previous work like [1],[2],[3] have fully discussed the importance of true few-shot setting for ICL. [4],[5],[6] also try to improve ICL under this setting. This setting is meaningful in practice.
>
> ***
>
> ## Q1:
> "The paper does not consider strong baselines.”
>
> ## Answer:
> “retrieval/ranking based on query similarity” is not a proper baseline because example selection or example order selection is not our work actually does. Besides, retrieval/ranking methods are orthogonal to our approach.
>
> Retrieval method needs a dataset to choose examples from, which contradicts our **true few-shot setting**.
>
>
> ***
>
> ## Q2:
> “The search algorithm proposed is computationally expensive”
>
> ## Answer:
> The computational complexity of our approach is not high at all.
> 1.The additional computation introduced by our method is a one-time cost, and we only need to search for proper weights one time before inference.
> 2.Beam search is linear in time complexity for the number of shots. Even for a 16-shot setting, searching for weights only needs 3*16 forward passes through language model.
> So the computational cost of our approach is totally acceptable.
>
>
> ***
>
> ## Q3:
> "In fig 6 a monotonic decrease is observed for the baseline ICL when number of demonstrations are increased"
>
> ## Answer:
>
> We have thoroughly examined our code and confirmed that there are no issues. In order to better align with the true few-shot setting, our experimental setup involves randomly selecting demonstrations, repeating the process 100 times and reporting the average performance. The best performance may increase with the number of shots, but it is not necessarily the same case when reporting average performance. Similar data of worse performance for larger shot numbers have also been observed in other ICL studies, such as [7].
>
> ***
>
> ## Q4:
> "line 051 posits that prior example selection strategies need a lot of data to work. However, this is not empirically validated. I would suggest still trying them out and showing the empirical performance. "
>
>
> ## Answer:
> Previous example selection strategies (e.g. [8],[9],[10]) need the whole training set or subset of original training to select examples from. For instance, [8] select examples from a candidate set of 100 or 1000 examples. However, this is contradictory to the true few-shot setting.
> Showing previous example selection performance as baselines is not suitable, because example selection is not the topic of our work. our work improves ICL performance by reweighting examples in a given demonstration. Besides, our work is under true few-shot setting, which also differs from example selection.
>
> ***
>
> ## Q5:
> "Is the search algorithm proposed sensitive to example order?"
>
> ## Answer:
> In our main experiment, each result represents the average result of 100 repeated trials with randomly sampled examples. This is sufficient to demonstrate the robustness of our method to order.
> Besides, we also observed that examples in the middle tend to be assigned less weight, while those at the beginning and end are assigned more.
>
> ***
>
> ## Q6:
> "How is the discretized weight space chosen?"
>
> ## Answer:
> Discretized weight space can be viewed as a hyper-parameter. We first try [0.8, 1.0, 1.2] and [0.9, 1.0, 1.1] on SST2, finding that [0.9, 1.0, 1.1] performs well and each weight is relatively balanced. So we choose [0.9, 1.0, 1.1] as the weight space.
>
> ***
>
> ## References
> [1]. True Few-Shot Learning with Language Models (Perez et al., NeurIPS 2021)
> [2]. True Few-Shot Learning with Prompts—A Real-World Perspective (Schick & Schütze, TACL 2022)
> [3]. FLEX: Unifying Evaluation for Few-Shot NLP (Bragg et al., NeurIPS 2021)
> [4]. Fantastically Ordered Prompts and Where to Find Them: Overcoming Few-Shot Prompt Order Sensitivity (Lu et al., ACL 2022)
> [5]. Noisy Channel Language Model Prompting for Few-Shot Text Classification (Min et al., ACL 2022)
> [6]. Z-ICL: Zero-Shot In-Context Learning with Pseudo-Demonstrations (Lyu et al., ACL 2023)
> [7]. Calibrate Before Use: Improving Few-shot Performance of Language Models (Zhao et al, ICML 2021)
> [8]. Active Example Selection for In-Context Learning (Zhang et al., EMNLP 2022)
> [9]. Self-Adaptive In-Context Learning: An Information Compression Perspective for In-Context Example Selection and Ordering (Wu et al., ACL 2023)
> [10]. What Makes Good In-Context Examples for GPT-3? (Liu et al., DeeLIO 2022)

---

### Official Review · Reviewer_FRfS · 2023-08-04

**Soundness:** 3

**Excitement:**

4: Strong: This paper deepens the understanding of some phenomenon or lowers the barriers to an existing research direction.

**Paper Topic And Main Contributions:**

This paper studies an interesting topic, which assigns different weights for different demonstrations in in-context learning. This paper argues that existing ICL methods treat all demonstrations equally and this could be further improved. This paper employs experiments to verify the importance of the weight of demonstrations. To obtain optimal weights, this paper proposed a masked self-prediction (MSP) score and employs beam search in the weight-searching process. And this paper further proposes two methods for applying these weights: scaling key matrix and scaling attention weights. Extensive experimental results show the effectiveness of the proposed method.

**Questions For The Authors:**

1. Rewighting demonstrations is orthogonal to demonstration selection.  And this paper conduct experiments on randomly selected demonstrations. Can reweighting demonstrations that are selected by existing SOTA selecting methods such as KATE and EPR further improve performance? Whether these well-selected demonstrations keep similar weights?
2. Existing studies show that ICL is sensitivity to the order. Does reweighting correlate with order? If we change the order, does the weight should be modified? If not, does reweighting reduce the sensitivity of order?
3. MSP does not require an extra validation set. An intuitive method that evaluates the quality of an example is using this example to conduct ICL and compare it with the zero-shot performance. With a random validation set, MSP, and this method, which is better?
4. Experiments are conducted on 8-shot and 16-shot settings, smaller shots, such as 2,3 are the proposed method still useful? Intuitively, reweighting seems to not work for smaller shots.

**Reasons To Accept:**

1. This paper finds a new problem of in-context learning: existing ICL methods treat demonstrations equally. And this paper argues that reweighting the demonstrations could further improve ICL and conducts experiments to show that the ICL performance varies significantly with different weights.
2. To find optimal weights, this paper proposes masked self-prediction score (MSP), which is correlated with the model's performance. And MSP does not require an additional validation set. And this paper employs beam search in the weight selection process. Then, this paper applies the weights by scaling key matrix and scaling attention weights. Section 5.4 also verifies that the proposed method outpeforms approximately 80% weights, which show its effectiveness.
3. Extensive experiments on 8 datasets across 5 LMs show the effectiveness of the proposed method.

**Reasons To Reject:**

1. Lack of more baselines. In Table 1, this paper only compares the proposed method with the original ICL. Only one baseline could not show the effectiveness of the proposed method. It is necessary to design other baselines which assign weights to different weights in ICL. For example, 1) assign random weights, 2) select the best weights from randomly assign k times. In specific, this paper should answer two main questions: 1) Why do we need to assign weights? and 2) Are the weights obtained using the proposed method better than other weights?
The experiments neglect the second question.
2. Lack of some heuristic conclusions. Such as, what types of demonstration are assigned higher/lower weights? Correlate this with label distributions, word distributions would benefit the total community.

**Reproducibility:**

3: Could reproduce the results with some difficulty. The settings of parameters are underspecified or subjectively determined; the training/evaluation data are not widely available.

**Reviewer Confidence:**

4: Quite sure. I tried to check the important points carefully. It's unlikely, though conceivable, that I missed something that should affect my ratings.

---

> ### Author Rebuttal · Authors · 2023-08-28
>
> # Responses to Reviewer FRfS
> Thank you for taking the time to thoroughly review our paper, and I sincerely appreciate your feedback. However, I want to clarify some of reasons for rejection that you've provided. The following are the responses to your reasons for rejection and questions.
>
> ## Q1:
> "Lack of more baselines.”
>
> ## Answer:
> The two baseline approaches you suggested for comparison, namely "1) assigning random weights" and "2) selecting the best weights from k randomly assigned trials," serve primarily to answer “Are the weights obtained using the proposed method better than other weights?”. Actually, we have answered this question in Section 5.4.
> In this section, we describe our process of randomly selecting 50 sets of weights, evaluating their accuracy on the validation set, and subsequently constructing a cumulative distribution function that maps accuracy distribution across the weight space. Notably, our analysis reveals that the weights generated by our method outperform over 80% of the weights.
> It's worth noting that this achievement is particularly impressive given that our method doesn't rely on any additional data.
>
> ***
>
> ## Q2:
> "Lack of some heuristic conclusions.”
>
> ## Answer:
> There is a certain correlation between weights and the positioning of examples, as well as the quality of examples. For instance, examples positioned in the middle tend to be assigned less weight, while higher-quality examples tend to be assigned more weight. That’s a good question, and we will add this part to section 5. Actually, even without these heuristic conclusions, our approach can automatically compute the weights for demonstrations. This eliminates the need for manual observation of demonstration features and further human intervention.
>
> ***
>
> ## Q3:
> "Reweighting demonstrations is orthogonal to demonstration selection”. Combining example reweighting with example selection.
>
> ## Answer:
>
> It's improper to combine our approach with example selection, because our approach is conducted under true few-shot setting while example selection is not.
>
> In **true few-shot setting**, a held-out validation set for better performance is unavailable, and resource is limited to only a few examples. Previous work like [1],[2],[3] have fully discussed the importance of true few-shot setting for ICL. [4],[5],[6] also try to improve ICL under this setting. However, SOTA selecting methods such as KATE and EPR need a dataset to select examples from, which is contradictory to the true few-shot setting. Combining our example reweighting with example selection ruins our true few-shot setting.
>
> The strength of our method lies in its ability to enhance performance for any given demonstration without the need for additional data. This randomness in example selection is more practical in true few-shot setting. Therefore, combining our method with example selection is not quite suitable. If forcefully applied, it would result in an all-ones vector as the weight for performant demonstration, which is already balanced in weight.
>
> ***
>
> ## Q4:
> " Does reweighting correlate with order?"
>
> ## Answer:
> In our main experiment, each result represents the average result of 100 repeated trials with randomly sampled examples. This is sufficient to demonstrate the robustness of our method to order.
> When the order of examples is changed, the weight for example needs to be modified. We also observed that examples in the middle tend to be assigned less weight, while those at the beginning and end are assigned more.
>
> ***
>
> ## Q5:
> "An intuitive method that evaluates the quality of an example is using this example to conduct ICL”. “With a random validation set, MSP, and this method, which is better?"
>
> ## Answer:
>
> In our preliminary experiments, we explored using 1-shot performance to gauge the quality of examples. However, the 1-shot performance does not necessarily reflect the significance of that particular example within the entire demonstration. The result is that our MSP outperforms this method.
>
> Comparing our MSP with an additional random validation dataset is unfair, as the validation set disrupts true few-shot setting.
>
> ***
>
> ## Q6:
> "Intuitively, reweighting seems to not work for smaller shots."
>
> ## Answer:
> Our method has a weak effect when the shot number is less than 4, which has been shown in Figure 6. We have also pointed it out in the Limitations. The effect of example weighting at low shot number is worth further exploration.
>
> ## References
> [1]. True Few-Shot Learning with Language Models (Perez et al., NeurIPS 2021)
> [2]. True Few-Shot Learning with Prompts—A Real-World Perspective (Schick & Schütze, TACL 2022)
> [3]. FLEX: Unifying Evaluation for Few-Shot NLP (Bragg et al., NeurIPS 2021)
> [4]. Fantastically Ordered Prompts and Where to Find Them: Overcoming Few-Shot Prompt Order Sensitivity (Lu et al., ACL 2022)
> [5]. Noisy Channel Language Model Prompting for Few-Shot Text Classification (Min et al., ACL 2022)
> [6]. Z-ICL: Zero-Shot In-Context Learning with Pseudo-Demonstrations (Lyu et al., ACL 2023)

---

### Meta-Review · Area_Chair_8j84 · 2023-09-17

**Recommendation:** 4

**Metareview:**

All reviewers agree that the authors present a simple yet effective algorithm to reassess possibly disparate importance of few-shot examples. To determine optimal weights for better In-Context Learning, the proposed algorithm seamlessly utilizes Masked-Self Prediction (MSP) scores and beam search, subsequently differentiating the importance by rescaling attention weights. To elaborate the current version, nevertheless, take into consider the following valuable points from our reviewers.

1)	Clarify the setting and relationship with demonstration selection: The current paper presumes the absence of any held-out validation data. Such “true few-shot setting” must be better clarified on the main draft.
2)	Stronger baselines: Although using held-out data could overwhelm the effect of using MSP and may not be congruent with a true few-shot setting, it remains pertinent to enrich the paper with additional experimental results. This could involve reweighting with an additional validation set or with example retrieval.
3)	Additional analysis on ordering sensitivity and inference costs: It would be advantageous to illuminate the interplay between the weights and the orders of few-shot example. In addition, considering the method predominantly operates with a higher number of shots, real-time inference cost (beyond asymptotic complexity) becomes relevant. Kindly incorporate some analysis.
4)	Potential implications for generative tasks: It would be beneficial to include some insights or preliminary thoughts when we apply the proposed method to generative tasks.

---

### Decision · Program_Chairs · 2023-10-07

**Decision:**

Accept-Findings

**Comment:**

All reviewers agree that the authors present a simple yet effective algorithm to reassess possibly disparate importance of few-shot examples. To determine optimal weights for better In-Context Learning, the proposed algorithm seamlessly utilizes Masked-Self Prediction (MSP) scores and beam search, subsequently differentiating the importance by rescaling attention weights. To elaborate the current version, nevertheless, take into consider the following valuable points from our reviewers.

1)	Clarify the setting and relationship with demonstration selection: The current paper presumes the absence of any held-out validation data. Such “true few-shot setting” must be better clarified on the main draft.
2)	Stronger baselines: Although using held-out data could overwhelm the effect of using MSP and may not be congruent with a true few-shot setting, it remains pertinent to enrich the paper with additional experimental results. This could involve reweighting with an additional validation set or with example retrieval.
3)	Additional analysis on ordering sensitivity and inference costs: It would be advantageous to illuminate the interplay between the weights and the orders of few-shot example. In addition, considering the method predominantly operates with a higher number of shots, real-time inference cost (beyond asymptotic complexity) becomes relevant. Kindly incorporate some analysis.
4)	Potential implications for generative tasks: It would be beneficial to include some insights or preliminary thoughts when we apply the proposed method to generative tasks.